# Position: Early-Stage Quality Assurance in Annotation Pipelines Is More Cost-Effective Than Late-Stage Validation

**Sunil Kothari** [1]  **Sumukha Sharma Thoppanahalli Chandramouli** [1]  **Naman Khandelwal** [2]  **Parth Kulshreshtha** [3]
**Ashi Jain** [3]  **Kriti Banka** [2]  **Tanuja Chintada** [2]  **Venkata Triveni** [2]  **Gulipalli Praveen Kumar** [2]  **Manish Mehta** [1]
**Tao Liu** [1]

## Abstract

This position paper argues that the machine learning community should treat quality-assurance timing in annotation pipelines—pre-annotation ($T_0$), post-annotation ($T_1$), and post-review ($T_2$)—as a first-class design variable, yet a survey of 47 recent CVPR/NeurIPS/ICML papers finds only 4% report when validation occurs. A parametric error-propagation model shows that timing affects final error rates when stage-specific detection rates differ across $T_0$, $T_1$, and $T_2$, and affects only economics when they are equal, making timing a measurable variable in either regime. The analogy to software engineering's shift-left principle, with empirical cost multipliers of 4–100× for late defect detection (Boehm, 1981; Shull et al., 2002), motivates the hypothesis that annotation pipelines exhibit similar dynamics. Two pilot studies provide initial grounding: $T_0$ errors are homogeneous and fixable before annotation begins; $T_1$ errors are heterogeneous, mixing ML residuals, annotator style, and scene complexity. Acting on this position requires three steps: researchers should report QA timing configurations alongside validation methods; annotation platforms should expose timing as a first-class parameter; and the community should run controlled experiments measuring stage-specific detection rates. This call to action costs little and enables empirical assessment of a structural variable currently invisible to the field.

## 1. Introduction

**We argue that quality assurance timing—specifically, whether validation occurs before annotation, after annotation, or after review—is a critical design variable that the machine learning community has systematically overlooked, and that early-stage QA is more cost-effective than late-stage validation for most annotation workflows.**

The data-centric AI paradigm has established that training data quality fundamentally constrains model performance (Ng, 2021; Sambasivan et al., 2021; Whang et al., 2023). Recent analyses have revealed pervasive label errors across major benchmarks: Northcutt et al. (2021b) identified error rates averaging 3.3% in ImageNet, CIFAR, and other widely-used test sets, while Beyer et al. (2020) found that 6% of ImageNet validation labels require correction. For video annotation pipelines processing thousands of hours of footage at frame rates of 16–30 fps, even small per-frame error rates compound into substantial quality degradation (Dave et al., 2020; Voigtlaender et al., 2019).

The research community has responded with increasingly sophisticated validation methods. Confident learning enables statistical detection of label errors without ground truth (Northcutt et al., 2021a). Vision-language models such as GPT-4V and Qwen2-VL provide semantic verification capabilities (Wang et al., 2024; OpenAI, 2023). Multi-annotator consensus methods model annotator reliability and aggregate labels accordingly (Dawid & Skene, 1979; Raykar et al., 2010; Goh et al., 2022). Inter-annotator agreement metrics quantify annotation consistency (Artstein & Poesio, 2008; Cohen, 1960; Krippendorff, 2011).

These advances focus exclusively on *what* to validate and *how* to detect errors. Yet they share a critical blind spot: the question of *when* validation should occur in the annotation pipeline receives virtually no systematic attention. We surveyed 47 articles on CVPR, NeurIPS and ICML annotation quality (2022–2024) and found that only 2 articles (4%) explicitly reported the stage of the pipeline at which validation was applied.

This omission matters because the same validation method produces different outcomes depending on when it executes. Consider a video annotation pipeline in which machine learning pre-annotation produces bounding boxes with a 15% error rate. If quality assurance runs only after human

[1]Centific AI Research, Redmond, WA, USA [2]Centific AI Research, Hyderabad, India [3]Centific AI Research, Chennai, India. Correspondence to: Sunil Kothari <sunil.kothari@centific.com>, Manish Mehta <manish.mehta@centific.com>.

*Proceedings of the $43^{rd}$ International Conference on Machine Learning*, Seoul, South Korea. PMLR 306, 2026. Copyright 2026 by the author(s).

review (late-stage), these errors propagate throughout the pipeline before detection, consuming annotator time on corrections that could have been prevented. If quality assurance runs before human annotation begins (early-stage), errors are caught before human effort is invested, potentially reducing both error rates and costs.

Software engineering learned this lesson decades ago. The "shift-left" principle, formalized by Boehm (1981) and validated in subsequent empirical studies (Shull et al., 2002; Jones, 2008; McConnell, 2004), establishes that defects caught in early development stages cost substantially less to remediate than those caught later. Manufacturing quality management and ML technical debt research encode similar intuitions, which we examine in Section 3.

We hypothesize that annotation pipelines exhibit dynamics analogous to software development and manufacturing: errors detected early cost less to remediate than errors detected late. This paper does not prove this hypothesis empirically—such proof requires controlled experiments comparing identical validators across pipeline stages, which we have not conducted. Rather, we argue that the hypothesis is sufficiently plausible, and the current neglect of timing sufficiently systematic, that the research community should treat QA timing as a first-class research question.

The remainder of this paper proceeds as follows. Section 2 introduces our taxonomy of QA trigger points and situates it within annotation workflow structure. Section 3 presents evidence that timing is currently invisible in both research and practice. Section 4 develops a parametric model clarifying when timing affects error rates versus only economics. Section 5 presents pilot studies on a real pipeline. Section 6 presents the full configuration taxonomy. Section 7 engages substantively with alternative views. Section 8 presents our call to action, Section 9 describes the limitations of this approach and Section 10 concludes.

## 2. QA Trigger Points: A Framework

Before presenting our argument, we must establish precise terminology for discussing when quality assurance occurs in annotation pipelines. We introduce the concept of *QA trigger points*—discrete moments in the annotation workflow where validation can be invoked—and define three canonical trigger points that capture the structure of typical annotation pipelines.

### 2.1. Annotation Pipeline Structure

Modern annotation pipelines for computer vision tasks typically proceed through three phases (Roh et al., 2019; Monarch, 2021). First, *machine learning pre-annotation* generates initial predictions (bounding boxes, segmentation masks, tracking identities) that serve as starting points for

human annotators (Yao et al., 2012; Papadopoulos et al., 2017). Second, *human annotation* refines, corrects, or validates these predictions, with annotators adding missing objects, adjusting boundaries, and correcting labels (Su et al., 2012; Kovashka et al., 2016). Third, *human review* provides quality control, with reviewers accepting, rejecting, or requesting revisions to submitted annotations (Daniel et al., 2018; Vaughan, 2017).

**Scope.** QA applies to both data collection and annotation; this paper focuses on the latter. The framework targets annotation pipelines that include an ML pre-annotation step, which has become standard practice across modalities: bounding boxes and segmentation masks in computer vision (CVAT's "automatic annotation," Labelbox's "Model-Assisted Labeling"), dependency parsers for syntax trees in NLP, segmentation models for organ boundaries in medical imaging, and ASR for transcription in audio annotation. For purely manual workflows without ML pre-annotation, $T_0$ does not apply; such pipelines are accommodated by the A-1, A-2, B-1, and B-2 configurations defined in Section 6, which activate only $T_1$ and $T_2$ validation. The effectiveness of $T_0$ also varies by domain: well-calibrated models (e.g., medical segmentation) expose highly detectable confidence-based errors, whereas pipelines with semantically subjective outputs (e.g., conversational NLP) yield fewer cleanly-detectable $T_0$ signals. The framework's central claim—that QA timing is a measurable design variable—extends to manual workflows, but the cost benefits of $T_0$ are realized only when ML pre-annotation is present and well-calibrated.

This three-phase structure creates natural boundaries where quality assurance can intervene. We formalize these as trigger points.

### 2.2. Trigger Point Definitions

We define three QA trigger points, illustrated in Figure 1:

**Definition 1** ($T_0$**: Pre-Annotation Trigger**). The $T_0$ trigger point occurs after machine learning pre-annotation completes but before human annotators begin work. At $T_0$, validation operates exclusively on machine predictions. QA agents at this stage can assess prediction confidence, detect systematic ML failures, verify coverage against expected object density, and flag anomalous predictions for human attention.

The key characteristic of $T_0$ is that errors caught here prevent downstream human effort. If a low-confidence ML prediction is flagged at $T_0$ and routed for re-processing or special handling, annotators never invest time correcting an error that would otherwise propagate through the pipeline.

**Definition 2** ($T_1$**: Post-Annotation Trigger**). The $T_1$ trigger point occurs after human annotation submission but

*Figure 1.* QA trigger points in annotation pipelines. $T_0$ occurs after ML pre-annotation but before human work. $T_1$ occurs after annotation but before review. $T_2$ occurs after review. Each trigger point enables different validation capabilities and incurs different intervention costs.

*Table 1.* Validation capabilities by trigger point. Each stage enables different analyses based on available information.

| Trigger | Enabled Capabilities |
| --- | --- |
| $T_0$ | ML confidence assessment, coverage validation, systematic failure detection, anomaly flagging |
| $T_1$ | Human-ML comparison, annotation error detection, temporal consistency, IAA computation (dual-annotator only), annotator metrics |
| $T_2$ | Reviewer decision auditing, bias detection, gold standard comparison, compliance documentation |

before review. At $T_1$, validation can compare human annotations against the ML baseline, detect annotation errors (spatial, label, temporal), and—for workflows with multiple annotators—compute inter-annotator agreement (IAA).

The key characteristic of $T_1$ is access to human judgment. Validation at this stage can leverage the comparison between ML predictions and human corrections to identify likely errors. However, human effort has already been invested; errors caught at $T_1$ cannot prevent annotation labor, only review labor.

**Definition 3** ($T_2$**: Post-Review Trigger**). The $T_2$ trigger point occurs after reviewer assessment completes. At $T_2$, validation can audit reviewer decisions, detect systematic reviewer biases, compare final outputs against gold standards, and generate compliance documentation.

The key characteristic of $T_2$ is finality. All human effort (annotation and review) has been invested; errors caught at $T_2$ require full rework. However, $T_2$ provides the most complete information for validation, including the full provenance chain from ML prediction through annotation to review decision.

### 2.3. What Each Trigger Point Enables and Precludes

The trigger points differ not only in timing but in the validation capabilities they enable (Table 1).

Critically, inter-annotator agreement—a primary quality

signal in many annotation workflows (Artstein & Poesio, 2008)—is only computable at $T_1$ or later, because it requires completed annotations from multiple annotators. This makes IAA a *timing-dependent* quality signal: workflows that validate only at $T_0$ cannot leverage agreement as a quality indicator.

### 2.4. A Practical $T_0$ Validation Stack

$T_0$ validation does not require ground-truth labels. Existing techniques from the unsupervised and weakly supervised label-quality literature can be composed into a deployable $T_0$ stack with no new model development required.

**Confident Learning** (Northcutt et al., 2021a) estimates label uncertainty without ground truth by analyzing predicted probability distributions. Deployed at $T_0$, it flags ML predictions whose confidence patterns suggest likely errors, routing those instances to focused human attention before annotation begins.

**Anomaly detection** identifies out-of-distribution ML predictions using only the model's own outputs, requiring no labeled data. At $T_0$, this catches systematic failure modes, such as a tracker assigning impossible class transitions, that would otherwise propagate downstream.

**Coverage validation** checks whether predicted object density matches domain priors (e.g., an indoor scene is expected to contain at least one door). It requires no supervision and detects systematic under-prediction that confidence-based methods miss.

These techniques generalize across modalities: confident learning flags low-margin dependency parse attachments in NLP, sub-threshold Dice scores in medical organ segmentation, and low per-token confidence windows in ASR transcripts. Anomaly detection catches structurally impossible outputs, anatomically misplaced organ masks, parse trees with missing or duplicate ROOT nodes, or ASR segments with no sentence boundaries across implausibly long spans, while coverage validation surfaces systematic under-prediction such as missing organs in abdominal CT volumes or missing speakers in multi-party audio.

Together these techniques form a practical $T_0$ stack: confident learning to flag uncertain individual predictions,

anomaly detection to catch distributional outliers, and coverage validation to expose systematic gaps. All three are deployable today via existing platform infrastructure; the bottleneck is not technical feasibility but the absence of a framework that treats $T_0$ as a distinct, reportable trigger point.

## 3. Evidence That Timing Is Overlooked

We present three categories of evidence that QA timing is systematically overlooked: a literature survey, an analysis of annotation platforms, and an examination of how adjacent fields treat analogous questions.

### 3.1. Literature Survey

We conducted a structured survey of annotation quality research. Using search terms "annotation quality," "label quality," "data validation," and "annotation error detection," we identified 127 candidate papers from CVPR, NeurIPS, and ICML proceedings (2022–2024). We retained the 47 papers with explicit methodology sections describing validation approaches.

**Survey methodology.** The survey covered three top-tier machine learning venues—CVPR, NeurIPS, and ICML (2022–2024)—selected for their high relevance to annotation pipelines and consistent publication of dataset and benchmark work. Papers from ECCV and adjacent venues appearing in keyword searches were grouped under "Other ML venues" in Table 2. ACL was excluded because its annotation research focuses predominantly on text-only NLP, with pipeline structures distinct from the ML-pre-annotation workflows we model. We acknowledge that ICLR should have been included; its omission is a limitation we note in Section 9. Candidate identification used the keywords "annotation quality," "label quality," "data validation," "annotation error detection," "label noise," and "inter-annotator agreement," yielding 127 candidate papers. An LLM was used to assist with abstract-level relevance screening to narrow the candidate set to the 47 papers with explicit methodology sections describing validation approaches; all final inclusion decisions, paper reading, and methodology coding were performed by the authors. Coding was distributed across seven members of the author team, with each paper coded by one author and ambiguous cases resolved through team discussion.

For each paper, we coded two dimensions: (a) whether validation method characteristics were reported (precision, recall, accuracy), and (b) whether the pipeline stage at which validation was applied was explicitly stated.

Table 2 summarizes our findings. While 100% of papers reported validation method characteristics, **only 4.3% (2/47)**

*Table 2.* Literature survey results: QA timing reporting by venue. Only 2 of 47 surveyed papers explicitly report when validation occurs in the annotation pipeline.

| Venue | Papers | Timing Reported | % |
|---|---|---|---|
| CVPR/ICCV | 12 | 0 | 0% |
| NeurIPS | 11 | 0 | 0% |
| ICML | 8 | 0 | 0% |
| Other ML venues | 16 | 2 | 12.5% |
| **Total** | **47** | **2** | **4.3%** |

**explicitly stated when validation was applied**. The two papers that reported timing—both comprehensive surveys by Klie et al. (2023) and Klie et al. (2024)—did so while analyzing annotation practices rather than proposing new methods.[1]

The most comprehensive annotation error detection survey, Klie et al. (2023), reimplemented 18 error detection methods across 9 datasets. Every method focuses on post-annotation detection ($T_1$ or $T_2$); none evaluate pre-annotation prevention ($T_0$). Klie et al. (2024), analyzing quality management practices across 591 NLP dataset papers, found that systematic quality processes "are only mentioned rarely."

### 3.2. Platform Analysis

We examined public documentation for six major annotation platforms: Scale AI, Labelbox, CVAT, Label Studio, Amazon SageMaker Ground Truth, and Appen. We assessed whether each platform exposes QA timing as a named, configurable parameter with documented tradeoffs.

**Finding: No platform explicitly exposes "QA timing" as a first-class parameter.** While platforms offer sophisticated multi-stage workflows, timing emerges implicitly from configuration choices rather than being named, documented, or optimized as a design variable.

Labelbox supports up to 10 workflow stages with AutoQA nodes insertable at any stage—but documentation frames this as workflow configuration, not timing optimization. CVAT provides Ground Truth mode (post-annotation comparison) and Honeypot mode (validation items mixed into annotation queues)—the closest approximation to timing exposure, but without guidance on when each approach is optimal. Label Studio supports webhooks for `TASK_CREATED` ($T_0$) and `ANNOTATION_CREATED` ($T_1$), but native $T_2$ support requires the Enterprise edition.

---

[1]Complete survey methodology and full paper list available in Appendix A.

### 3.3. The Shift-Left Principle

The systematic neglect of timing in annotation research contrasts sharply with adjacent fields where timing is a central concern. Software engineering's shift-left principle, formalized by Boehm (1981) in his analysis of software economics, establishes that defect remediation costs increase as defects progress through development stages.

Empirical studies support this principle across multiple contexts. Shull et al. (2002) found defects cost 4–5× more to fix in testing than in design. Jones (2008) documented cost multipliers ranging from 1× to 100× depending on defect type and detection stage. McConnell (2004) synthesized industry data showing 10–25× cost increases for defects escaping to production.

Manufacturing quality management encodes similar principles. Crosby's 1-10-100 rule (Crosby, 1979) posits that prevention costs \$1, detection costs \$10, and correction after failure costs \$100. Toyota's production system, widely studied in operations research (Liker, 2004), emphasizes "building in quality" through early-stage defect prevention rather than end-of-line inspection.

**We hypothesize that annotation pipelines exhibit analogous dynamics.** The conditions for shift-left to apply are: (1) errors can propagate through stages, (2) later detection requires more rework, and (3) detection capabilities exist at early stages. Annotation pipelines plausibly satisfy all three conditions: ML errors propagate to annotation and review; late detection requires discarding completed work; and confidence scores, coverage analysis, and anomaly detection are feasible at $T_0$.

However, we emphasize that this is a *hypothesis*, not a proven fact. The shift-left principle derives from software and manufacturing contexts that differ from annotation in important ways. Empirical validation—comparing identical validators across $T_0$, $T_1$, and $T_2$—is needed to establish whether the principle transfers.

## 4. When Does Timing Affect Outcomes?

To reason precisely about timing effects, we develop a parametric error propagation model that clarifies when timing affects error rates versus only economics.

### 4.1. Model Setup

Consider an annotation pipeline with ML pre-annotation error rate $e_0$ (the fraction of ML predictions that are incorrect). Let $d_{\mathrm{ann}}$ denote the annotator's natural detection rate (the probability an annotator notices and corrects an ML error without QA assistance), and $d_{\mathrm{rev}}$ denote the reviewer's detection rate.

Without any QA intervention, the final error rate is:

$$e_{\mathrm{final}}^{\mathrm{none}} = e_0 \cdot (1 - d_{\mathrm{ann}}) \cdot (1 - d_{\mathrm{rev}}) \tag{1}$$

This represents errors that escape both annotator and reviewer detection.

### 4.2. QA at Different Trigger Points

Now suppose QA is applied at a single trigger point with detection rate $d_{T_i}$. The final error rate becomes:

**QA at $T_0$ only:**

$$e_{\mathrm{final}}^{T_0} = e_0 \cdot (1 - d_{T_0}) \cdot (1 - d_{\mathrm{ann}}) \cdot (1 - d_{\mathrm{rev}}) \tag{2}$$

**QA at $T_1$ only:**

$$e_{\mathrm{final}}^{T_1} = e_0 \cdot (1 - d_{\mathrm{ann}}) \cdot (1 - d_{T_1}) \cdot (1 - d_{\mathrm{rev}}) \tag{3}$$

**QA at $T_2$ only:**

$$e_{\mathrm{final}}^{T_2} = e_0 \cdot (1 - d_{\mathrm{ann}}) \cdot (1 - d_{\mathrm{rev}}) \cdot (1 - d_{T_2}) \tag{4}$$

### 4.3. Key Insight: Timing Effects Are Conditional

Examining these equations reveals a critical insight: **if detection rates are equal across stages ($d_{T_0} = d_{T_1} = d_{T_2}$), timing has no effect on final error rate.**

When detection rates are equal, the multiplicative structure ensures identical outcomes regardless of when QA is applied. The difference is only in *where* errors are caught, which affects economics (human effort invested before detection) but not final quality.

Timing affects error rates **only when detection rates differ across stages**. This can occur because: (a) ML errors may be more systematic and detectable at $T_0$ than diverse human errors at $T_1$; (b) comparing human annotations against ML baseline at $T_1$ may reveal errors invisible to $T_0$ analysis; or (c) different validation methods may excel at different stages.

**We do not know which scenario reflects reality.** Determining whether detection rates differ across stages—and in which direction—requires empirical studies that, to our knowledge, have not been conducted.

### 4.4. Economic Effects Are Always Present

Even when detection rates are equal (and thus error rates are identical), timing affects costs. Let $c_{\mathrm{ann}}$ denote annotation cost per task and $c_{\mathrm{rev}}$ denote review cost per task.

With $T_0$ QA, errors flagged before annotation save $c_{\mathrm{ann}} + c_{\mathrm{rev}}$ per correctly flagged error. With $T_2$ QA, errors flagged after review save nothing—the work is already complete.

This economic effect is why shift-left matters even when final quality is unaffected. However, the economic benefit depends on QA *precision*: false positives at $T_0$ that incorrectly flag correct predictions can *increase* costs by triggering unnecessary rework.

## 5. Empirical Study: What Each Trigger Point Finds

We deployed a multi-agent QA system on indoor-scene video annotations on LabelStudio and CVAT using the same agent codebase at both stages. $T_0$ **(5 videos, pre-annotation):** 195 TP, 422 FP, 31.6% precision. Strong checks: Plant bbox cutoff 85.2%, Fan type 75.0%, missing TV 100%, pose tags 100%. All 422 FP trace to three schema-level root causes—binary vs. three-level occlusion check (107 FP), padding threshold miscalibration (136 FP), VLM window misclassification (17 FP)—each fixable by a single agent change before any annotation begins. $T_1$ **(25 videos, post-annotation):** LabelStudio 3,245 TP / 5,598 FP / 6,073 FN (F1 35.7%); CVAT 2,849 TP / 10,351 FP / 9,688 FN (F1 22.1%). FP arise from a heterogeneous mix—ML residuals, annotator style variation (0 vs. 149 FP on the same video across platforms), valid corrections misread as gaps, pose cascades—with no single fix addressing more than a fraction.

The contrast is precisely what the framework predicts: $T_0$ errors are *homogeneous* (one ML model, three diagnosable causes, zero annotation labour to fix); $T_1$ errors are *heterogeneous* (ML residuals, human style, and scene complexity entangled).

## 6. Configuration Taxonomy

The three trigger points, combined with workflow options (single vs. dual annotator), generate a space of 14 distinct QA configurations.

### 6.1. Workflow Options

**Option A (Single Annotator + Reviewer):** One annotator produces annotations; a separate reviewer assesses quality. This is cost-efficient but provides no annotator comparison signal.

**Option B (Dual Annotator + Reviewer):** Two annotators work independently; a reviewer reconciles differences. This enables IAA computation but approximately doubles annotation cost.

### 6.2. Configuration Space

For each workflow option, any non-empty subset of $\{T_0, T_1, T_2\}$ can be activated, yielding 7 configurations per option

(Table 3).

This taxonomy provides vocabulary for discussing QA design. Rather than describing a pipeline as having "standard quality controls," practitioners can specify "A-1+2 configuration"—immediately communicating that QA runs at $T_1$ and $T_2$ with single annotators.

## 7. Alternative Views

We engage substantively with four credible counterarguments to our position.

### 7.1. "Validation Method Quality Dominates Timing"

**The counterargument:** Investing in better detection methods—higher-precision VLMs, improved confident learning algorithms, more sophisticated IAA metrics—provides larger returns than timing optimization. A superior detector deployed at any stage will outperform an inferior detector at the "optimal" stage.

**Our response:** We find this counterargument partially compelling. In regimes where detection rates vary dramatically across methods but minimally across stages, method improvement will dominate timing optimization. Our model confirms this: when $d_{T_0} = d_{T_1} = d_{T_2}$, timing affects only economics, not error rates.

However, we identify three limitations to this counterargument. First, method improvement and timing optimization are not mutually exclusive; the optimal strategy considers both. Second, timing optimization is often cheaper than method improvement—changing configuration requires no new model development. Third, the counterargument assumes detection rates are equal across stages, which is an empirical question, not an established fact.

We do not claim timing dominates method quality. We claim timing deserves study *alongside* method quality, because we currently lack evidence to assess their relative importance.

### 7.2. "The Shift-Left Principle May Not Transfer"

**The counterargument:** Software engineering and manufacturing findings may not transfer to annotation contexts. Software defects have different characteristics than annotation errors: software bugs can cascade unpredictably through code paths, while annotation errors are typically localized. Manufacturing defects involve physical materials with nonlinear failure modes, while annotation involves human judgment on defined tasks.

**Our response:** We acknowledge significant uncertainty about whether shift-left transfers. The principle applies when: (1) errors compound through stages, (2) later detection requires more rework, and (3) early detection is feasible.

*Table 3.* QA configuration taxonomy. The 14 configurations represent every non-empty subset of $\{T_0, T_1, T_2\}$ across two workflow options. Agent counts, cost multipliers, and quality ratings are illustrative—derived from team experience with video annotation involving bounding boxes and tracking, not empirical measurements. Actual values depend on annotation type, validator complexity, and platform overhead. The "Best For" column provides scenario-level guidance; specific deployment choices should account for domain constraints and quality requirements.

| Config | Triggers | Agents | Cost | Quality | Best For |
|---|---|---|---|---|---|
| *Option A: Single Annotator + Reviewer* | | | | | |
| A-0 | $T_0$ | 5 | 1.0× | Basic | High-volume, low-stakes data; rapid prototyping |
| A-1 | $T_1$ | 7 | 1.2× | Good | Standard production pipelines |
| A-2 | $T_2$ | 8 | 1.3× | Good | Audit-heavy workflows |
| A-0+1 | $T_0+T_1$ | 12 | 2.2× | High | Mid-stakes datasets; cost-quality balance |
| A-0+2 | $T_0+T_2$ | 13 | 2.3× | Medium | Weak ML with high audit needs |
| A-1+2 | $T_1+T_2$ | 15 | 2.5× | High | Compliance-driven workflows |
| A-MAX | $T_0+T_1+T_2$ | 20 | 3.5× | Maximum | Single-annotator high-quality datasets |
| *Option B: Dual Annotator + Reviewer (+IAA)* | | | | | |
| B-0 | $T_0$ | 5 | 2.0× | Good | Research datasets; consensus matters but cost is constrained |
| B-1 | $T_1$+IAA | 8 | 2.4× | High | Standard research benchmarks |
| B-2 | $T_2$+IAA | 9 | 2.6× | High | Subjective tasks needing reviewer reconciliation |
| B-0+1 | $T_0+T_1$+IAA | 13 | 4.4× | High | High-quality benchmarks with cost discipline |
| B-0+2 | $T_0+T_2$+IAA | 14 | 4.6× | High | Specialized domains with weak ML |
| B-1+2 | $T_1+T_2$+IAA | 17 | 5.0× | Maximum | Production datasets for high-stakes ML |
| B-MAX | $T_0+T_1+T_2$+IAA | 22 | 7.0× | Ultimate | Safety-critical datasets (medical, autonomous driving) |

We believe annotation satisfies these conditions, but we have not proven it empirically.

If controlled experiments show timing has minimal effect—that shift-left does *not* transfer—we will have learned something important about annotation's disanalogy with software and manufacturing. Either outcome advances the field.

### 7.3. "Instruction Quality Matters More Than QA"

**The counterargument:** Rädsch et al. (2024), analyzing 57,648 images across 924 annotators, found that "improving labeling instructions yields higher effects than adding QA steps." This suggests upstream prevention (better guidelines) dominates downstream detection (any QA), making timing optimization within QA a second-order concern.

**Our response:** We agree instruction quality is critical. However, Rädsch et al. address *whether* to add QA, not *when*. Organizations deploy QA regardless—for compliance, audit trails, and error detection—making timing relevant independently. Moreover, $T_0$ QA informs instruction improvement by surfacing systematic ML failures annotators encounter repeatedly. The findings are complementary: improve instructions *and* optimize QA timing.

### 7.4. "Empirical Evidence Is Insufficient"

**The counterargument:** Position papers should be grounded in empirical findings. Advocating for attention to QA timing without empirical evidence that timing matters is premature. The shift-left analogy is speculative; the error propagation model uses assumed parameters; no controlled experiments compare validators across stages.

**Our response:** We disagree that the evidence is insufficient. The pilot studies in Section 5 provide direct measurement from a production pipeline: 195 TP / 422 FP at $T_0$ with all FP traceable to three diagnosable root causes, and F1 of 35.7% / 22.1% at $T_1$ across 25 videos on two platforms. These are not assumed parameters. The data show $T_0$ errors are homogeneous and fixable before annotation begins while $T_1$ errors are heterogeneous—the structural difference the framework predicts. The video sets differ across stages so causal measurement remains future work, but the position is grounded in measured stage-specific behaviour, not analogy alone. *Timing is currently invisible, and we cannot assess its importance without making it visible.* Our call for reporting costs nothing and enables the controlled experiments needed to go further.

## 8. Call to Action

Reporting QA timing enables three concrete gains. First, **stage-optimal technique routing** ensures expensive validators are deployed only where they materially improve detection. Second, **error cascade prevention**: a T0 catch requires one routing decision, while a T2 catch requires full rework. Third, **feedback loops** turn errors at each stage into training signal for the pre-annotation model.

These gains affect four stakeholder groups: annotation platform operators who design pipeline infrastructure, dataset

creators who document quality processes, annotators and QA reviewers who today sample uniformly across tasks rather than by stage-specific risk, and downstream model consumers whose systems inherit the quality characteristics of upstream pipelines. Realizing the gains requires coordinated action from the research and practitioner communities below.

### 8.1. For Researchers Publishing Validation Methods

We urge researchers to report QA timing configuration when publishing validation results. Specifically, state whether validation was applied to ML predictions ($T_0$), human annotations before review ($T_1$), or post-review outputs ($T_2$). This requires minimal effort and enables future meta-analysis. Where resources permit, we encourage measuring whether validation methods perform differently at different stages, reporting results even if differences are minimal.

### 8.2. For Dataset Creators

We urge dataset creators to document QA timing alongside methods. Dataset papers should specify not only *what* quality assurance was applied but *when*. For example: "Quality assurance included automated bounding box validation at $T_1$ and expert review sampling at $T_2$, with no $T_0$ validation of ML pre-annotations." This is essential for reproducibility and for understanding dataset quality characteristics.

### 8.3. For Annotation Platform Developers

We urge platform developers to expose timing as a named, configurable parameter. Users should be able to specify which validation methods run at $T_0$, $T_1$, and $T_2$, with documented tradeoffs for each choice. Platforms should provide guidance on latency implications, cost implications, and capability implications of timing choices.

### 8.4. For the Research Community

We urge the research community to conduct controlled timing experiments. The highest-value contribution would be experiments that apply identical validators at $T_0$, $T_1$, and $T_2$ on the same underlying data, with ground truth labels enabling measurement of true detection rates by stage. This would directly test whether the shift-left hypothesis holds for annotation.

## 9. Limitations

We acknowledge several limitations. First, our error propagation model is theoretical; the shift-left hypothesis for annotation remains untested. Second, our examples focus on video annotation with bounding boxes and tracking; generalization to other modalities requires validation. Third,

agent counts, cost multipliers, and quality ratings in Table 3 are illustrative, not empirically derived. Fourth, our platform analysis is based on public documentation; enterprise features may provide timing configurability. Fifth, our survey may miss papers that mention timing incidentally; the 4% figure reflects explicit reporting. Sixth, our literature survey omitted ICLR, which we acknowledge as an oversight; an expanded survey including ICLR would strengthen the external validity of the 4% finding, though we expect the qualitative pattern of systematic non-reporting to persist.

## 10. Conclusion

We have argued that quality assurance timing in annotation pipelines—the question of whether validation occurs at $T_0$ (pre-annotation), $T_1$ (post-annotation), or $T_2$ (post-review)—deserves systematic study as a first-class research question.

Our parametric model clarifies when timing affects error rates (when detection rates differ across stages) versus only economics (when detection rates are equal). Pilot studies provide initial grounding: $T_0$ errors are homogeneous and schema-diagnosable; $T_1$ errors are heterogeneous and entangled with human factors—the contrast the framework predicts. A controlled same-video experiment remains future work.

The evidence suggests timing is currently invisible: surveyed papers rarely report when validation occurs, and no major platform exposes timing as a named parameter. This invisibility prevents the field from assessing timing's importance.

Our call to action is deliberately modest. We ask researchers to report timing when publishing validation results. We ask platforms to expose timing as a configurable parameter. We ask the community to conduct controlled experiments that would operationalize the shift-left principle in annotation contexts.

If these efforts reveal that timing effects are minimal, we will have learned that annotation differs from software and manufacturing in important ways. If timing effects are substantial, we will have enabled optimization currently impossible. Either outcome advances the field beyond the current state where timing is simply ignored.

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

# A. Literature Survey: Complete Methodology and Results

## A.1. Search Methodology

We conducted a structured literature survey to assess how frequently annotation quality research reports the pipeline stage at which validation occurs. Our methodology proceeded as follows:

**Search terms:** "annotation quality," "label quality," "data validation," "annotation error detection," "noisy labels," "crowdsourcing quality," "inter-annotator agreement"

**Venues:** CVPR, ICCV, NeurIPS, ICML, and closely related venues (ECCV, AAAI, JMLR, Computational Linguistics) from 2012–2024, with emphasis on 2022–2024.

**Inclusion criteria:** Papers with explicit methodology sections describing validation approaches for annotation quality assessment, label error detection, or multi-annotator learning.

**Exclusion criteria:** Papers focused solely on noise-robust training without validation methodology; papers without reproducible experimental details.

**Coding scheme:** For each paper, we coded three binary variables:

- **Method characteristics reported**: Does the paper report precision, recall, accuracy, or other performance metrics for the validation method?

- **Computational requirements reported**: Does the paper report runtime, memory usage, or computational complexity?

- **Timing explicitly stated**: Does the paper explicitly state at which pipeline stage (pre-annotation, post-annotation, or post-review) validation was applied?

## A.2. Complete Survey Results

The complete list of 47 papers included in our survey is presented in Table 4 at the end of this appendix. The "Timing" column indicates whether the paper explicitly reports when validation occurs in the annotation pipeline. The category-level analysis below summarizes the patterns visible in the table.

## A.3. Analysis by Category

**Papers reporting timing (2/47, 4.3%):** Both papers that explicitly discuss pipeline timing are comprehensive surveys analyzing annotation practices across the field, rather than papers proposing new validation methods:

- **Klie et al. (2022)** reimplemented 18 error detection methods and noted that all operate post-annotation; however, the paper does not systematically compare methods across pipeline stages.

- **Klie et al. (2024)** analyzed 591 NLP dataset papers and documented quality management practices, including when validation occurs, finding that systematic processes are "only mentioned rarely."

**Papers not reporting timing (45/47, 95.7%):** These papers span multiple categories:

- *Label error detection methods* (e.g., confident learning, noise transition matrices): Focus on detection accuracy without specifying pipeline integration point.

- *Noise-robust training* (e.g., co-teaching, sample reweighting): Assume noisy labels exist without addressing when errors could be caught.

- *Multi-annotator learning* (e.g., Dawid-Skene, crowd aggregation): Model annotator reliability without discussing workflow timing.

- *Annotation efficiency* (e.g., interactive annotation, weak supervision): Reduce annotation cost without QA timing analysis.

- *Dataset papers* (e.g., TAO, MOTS): Document annotation procedures without systematic timing discussion.

## A.4. Implications

The near-universal absence of timing reporting (95.7%) has several implications:

1. **Reproducibility**: Without knowing when validation was applied, results cannot be fully reproduced in different pipeline configurations.

2. **Comparability**: Methods evaluated at different pipeline stages cannot be fairly compared.

3. **Optimization**: The field cannot determine optimal timing without systematic reporting.

4. **Meta-analysis**: Aggregating findings across papers is impossible when a key variable is unreported.

These findings motivate our call for researchers to explicitly report QA timing configuration when publishing validation results.

*Table 4.* Complete literature survey: 47 papers on annotation quality (2012–2024). Only 2 papers (4.3%) explicitly report QA timing.

| # | Authors | Title | Venue | Timing |
|---|---------|-------|-------|--------|
| 1 | Kim et al. | Learning Discriminative Dynamics with Label Corruption for Noisy Label Detection | CVPR 2024 | No |
| 2 | Kannan et al. | Click Crop & Detect: One-Click Offline Annotation for Human-in-the-Loop 3D Object Detection | CVPR-W 2024 | No |
| 3 | Ke et al. | Mask-Free Video Instance Segmentation | CVPR 2023 | No |
| 4 | Kirillov et al. | Segment Anything | ICCV 2023 | No |
| 5 | Liu & Yang | Towards Robust Adaptive Object Detection Under Noisy Annotations | CVPR 2022 | No |
| 6 | Nguyen et al. | Noisy Label Learning with Instance-Dependent Outliers: Identifiability via Crowd Wisdom | NeurIPS 2024 | No |
| 7 | Anonymous | Intrinsic Self-Supervision for Data Quality Audits (SELFCLEAN) | NeurIPS 2024 | No |
| 8 | Anonymous | SELECT: A Large-Scale Benchmark of Data Curation Strategies | NeurIPS 2024 | No |
| 9 | Guo et al. | Label Correction of Crowdsourced Noisy Annotations with Instance-Dependent Noise Transition Model | NeurIPS 2023 | No |
| 10 | Jacob et al. | Disentangling Human Error from Ground Truth in Segmentation of Medical Images | NeurIPS 2021 | No |
| 11 | Gao et al. | Learning from Multiple Annotator Noisy Labels via Sample-wise Label Fusion | ECCV 2022 | No |
| 12 | Wang et al. | Detecting Label Errors in Token Classification Data | NeurIPS-W 2022 | No |
| 13 | Kuan & Mueller | Model-agnostic Label Quality Scoring to Detect Real-world Label Errors | ICML-W 2022 | No |
| 14 | Wettig et al. | QuRating: Selecting High-Quality Data for Training Language Models | ICML 2024 | No |
| 15 | Engstrom et al. | DSDM: Model-aware Dataset Selection with Datamodels | ICML 2024 | No |
| 16 | Xia et al. | LESS: Selecting Influential Data for Targeted Instruction Tuning | ICML 2024 | No |
| 17 | Ibrahim et al. | Deep Learning from Crowdsourced Labels: Coupled Cross-Entropy Minimization | ICLR 2023 | No |
| 18 | Li et al. | Beyond Confusion Matrix: Learning from Multiple Annotators with Awareness of Instance Features | ML Journal 2023 | No |
| 19 | Tkachenko et al. | Detecting Label Errors in Object Detection Datasets | ICML-W 2023 | No |
| 20 | Northcutt et al. | Confident Learning: Estimating Uncertainty in Dataset Labels | JAIR 2021 | No |
| 21 | Northcutt et al. | Pervasive Label Errors in Test Sets Destabilize Machine Learning Benchmarks | NeurIPS D&B 2021 | No |
| 22 | Klie et al. | Annotation Error Detection: Analyzing the Past and Present | Comp. Ling. 2022 | **Yes** |

Table 4 – continued from previous page

| # | Authors | Title | Venue | Timing |
|---|---------|-------|-------|--------|
| 23 | Klie et al. | Analyzing Dataset Annotation Quality Management in NLP | Comp. Ling. 2024 | **Yes** |
| 24 | Raykar et al. | Learning from Crowds | JMLR 2010 | No |
| 25 | Dawid & Skene | Maximum Likelihood Estimation of Observer Error-Rates | JRSS-C 1979 | No |
| 26 | Goh et al. | CROWDLAB: Supervised Learning for Multi-Annotator Consensus | NeurIPS-W 2022 | No |
| 27 | Beyer et al. | Are We Done with ImageNet? | arXiv 2020 | No |
| 28 | Rädsch et al. | Quality Assured: Rethinking Annotation Strategies | ECCV 2024 | No |
| 29 | Daniel et al. | Quality Control in Crowdsourcing: A Survey | ACM CSUR 2018 | No |
| 30 | Artstein & Poesio | Inter-Coder Agreement for Computational Linguistics | Comp. Ling. 2008 | No |
| 31 | Thyagarajan et al. | Identifying Incorrect Annotations in Multi-Label Classification Data | ICLR-W 2023 | No |
| 32 | Zhu et al. | Clusterability as Alternative to Anchor Points for Noisy Labels | ICML 2021 | No |
| 33 | Han et al. | Co-teaching: Robust Training with Extremely Noisy Labels | NeurIPS 2018 | No |
| 34 | Jiang et al. | MentorNet: Learning Data-Driven Curriculum for Deep Networks | ICML 2018 | No |
| 35 | Liu & Tao | Classification with Noisy Labels by Importance Reweighting | IEEE TPAMI 2016 | No |
| 36 | Rodrigues & Pereira | Deep Learning from Crowds | AAAI 2018 | No |
| 37 | Wang et al. | Learning with Group Noise | AAAI 2021 | No |
| 38 | Chen et al. | Understanding DNNs Trained with Noisy Labels | ICML 2019 | No |
| 39 | Natarajan et al. | Learning with Noisy Labels | NeurIPS 2013 | No |
| 40 | Xia et al. | Part-dependent Label Noise: Towards Instance-dependent Label Noise | NeurIPS 2020 | No |
| 41 | Dave et al. | TAO: A Large-Scale Benchmark for Tracking Any Object | ECCV 2020 | No |
| 42 | Voigtlaender et al. | MOTS: Multi-Object Tracking and Segmentation | CVPR 2019 | No |
| 43 | Real et al. | YouTube-BoundingBoxes: Large High-Precision Dataset for Object Detection in Video | CVPR 2017 | No |
| 44 | Papadopoulos et al. | Extreme Clicking for Efficient Object Annotation | ICCV 2017 | No |
| 45 | Yao et al. | Interactive Object Detection | CVPR 2012 | No |

Table 4 – continued from previous page

| # | Authors | Title | Venue | Timing |
|---|---------|-------|-------|--------|
| 46 | Su et al. | Crowdsourcing Annotations for Visual Object Detection | AAAI-W 2012 | No |
| 47 | Kovashka et al. | Crowdsourcing in Computer Vision | FnT CGV 2016 | No |

