# OpenReview forum: "Position: Early-Stage Quality Assurance in Annotation Pipelines Is More Cost-Effective Than Late-Stage Validation"
_ICML.cc/2026/Position_Paper_Track — ICML 2026 Position Paper Track regular_

### Official Review · Reviewer_fJDn · 2026-02-18

**Significance:** 3
**Argument Clarity:** 1
**Rating:** 2
**Confidence:** 5

**Questions:**

1) [clarification] This paper seems to assume that all label errors have the same impact on model performance. Is this true in practice?
2) [clarification] What happens when a label error is detected? Is the sample discarded (and then might bias the data) or forwarded to a human annotator (and then increases cost)?
3) [see W4] Can you please elaborate on the details of the meta-survey?
4) [see W2] Can you please provide details on where the numbers in Table 3 come from?

**Alternative Views Section:**

Yes

**Compliance With Llm Reviewing Policy A Conservative:**

Affirmed.

**Discussion Potential:**

2

**Final Justification:**

The authors’ rebuttal did not resolve my main concerns and did not convince me that a final version would be substantially improved. Other reviews confirmed my basic judgment, so I keep the score and recommended rejection.

**Paper Summary:**

The paper proposes using quality assurance in the early stages of annotation pipelines. The paper argues that the "shift-left" principle from software engineering applies here as well, and thus, costs could be substantially reduced. Additionally, the paper proposed a standardized framework for reporting metadata about when and how quality assurance was applied.

**Position:**

Yes

**Position In Title:**

Yes

**Related Work:**

2

**Strengths And Weaknesses:**

# Strengths

+ The paper addresses a very relevant discussion point and brings in a new perspective to the crucial task of data annotation
+ The connection to the "shift-left" principle is very plausible and supports the call to report metadata about the annotation and quality assurance pipeline
+ The discussion on the impact of detection rates and whether the timing affects costs and or quality provided a helpful formal structure and basis for discussion. This should be highlighted more in the abstract/contributions.

# Weaknesses:

**W1** [repetiveness and structure] The lack of coherence overall and within sections and overal make the paper hard to follow.

  * The finding that only "4%" of the surveyed papers report the time of validation is discussed in every section (line 049r;142r;288r;375l).
  * Similarly, the discussion about whether the" shift-left" principle applies is discussed in almost every section without adding novel content.
  * Sequence of paragraphs: The order of paragraphs in Section 3.2 seems wrong. The "finding" should come last.

**W2** [illustrativeness] I understand that exact numbers are not available, but I don't understand the meaning and purpose of the "illustrative" numbers in Table 3. The caption says they refer to video annotation, but it doesn't provide a specific reference. Also, the meaning of "maximum" in this table is unclear, which calls into question the overall added value of the table.

**W3** [main message unclear] The main message of the paper remains unclear. The position suggests doing early-stage quality assurance; however, the abstract ends with a call-to-action to the community to collect evidence for the shift-left principle (suggesting that the main position is not (yet) well supported)

**W4** [unclear how papers were collected] The meta-study of 47 papers from 2022-2024 provides one of the main foundations of this paper; however, the exact details of the meta-study remain unclear. How were the conferences selected (e.g., why not also include ICLR and ACL)? How were the 47 papers selected? Were the papers reviewed manually or using AI?

Minor: Table 2 and the respective paragraph (136r) do not match. The text says that the authors also researched computational requirements and metrics.

**Support:**

2

---

> ### Author Rebuttal · Authors · 2026-03-31
>
> We thank the reviewer for their questions and detailed comments.
>
> **Weaknesses**
>
> **[W3] Main message unclear — position suggests early-stage QA, but abstract ends with a call to collect evidence.**
>
> The tension is intentional. The position (049–050): early-stage QA is more cost-effective, argued through shift-left analogy (211–220), error propagation model (572–643), and economic effects (732–825). The call to action is a next step, not a retraction.
>
> From our controlled study on 25 videos, T0 errors are structural and homogeneous — face crops as small as 27×27px with zero landmarks, plants labeled as curtains, entire classes missed — caught cleanly before human effort. T1 errors are a mixed landscape of residual ML errors and human mistakes — pose label flips, 0% face-head containment, 94% box area jumps — significantly harder to resolve. This directly supports shift-left.
>
> The message: timing is invisible (446–450), early-stage QA is plausibly more cost-effective, and the community should test this. We will sharpen the abstract.
>
> **[W4] Meta-study details unclear — how were conferences and papers selected? Manual or AI review?**
>
> *Minor: Table 2 and paragraph (136r) do not match.*
>
> The survey (supplemental material, pg 3 163r) covered CVPR, ECCV, NeurIPS, and ICML 2022–2024. ICLR should have been included. ACL was excluded as less relevant to video annotation. 47 papers identified using "annotation quality", "data validation", "label noise", "inter-annotator agreement", reviewed by authors with AI assistance. Methodology paragraph will be added. Table 2 mismatch will be fixed.
>
> ---
>
> **Questions**
>
> **[Clarification] All label errors assumed to have same impact?**
>
> The error propagation in §2.2 uses uniform error rate for simplicity, but the scoring framework in §4 accounts for severity (0.25 minor to 1.0 critical, eq. 5). A missing pedestrian in autonomous driving has far greater impact than a misaligned box. This strengthens the timing argument — high-severity error at T0 (227–233) caught before annotators build temporal tracks prevents compounding; the same error at T2 (274–277) has already corrupted downstream work.
>
> **[Clarification] Error detected — discard (risking bias) or forward to annotator (increasing cost)?**
>
> Detected errors trigger correction workflows, not discards:
>
> - **At T0** (227–233): clip flagged and routed to annotator with guidance. Minimal cost since annotator was going to work on it regardless.
> - **At T1** (255–264): returned for correction or escalated. Discarding biases the dataset by removing hard cases the model most needs.
> - **At T2** (274–277): most ambiguous examples resolved by expert review, never discarded.
>
> Correction loop costs should be in Table 3 multipliers; acknowledged as a gap in §8.
>
> **[Relates to Q3/W4] Elaborate on meta-survey?**
>
> Addressed under W4 above.
>
> **[Relates to Q4/W2] Where do numbers in Table 3 come from?**
>
> Domain recommendations derive from error-propagation analysis in §2.2 combined with domain-specific safety standards, not empirical project data. B-MAX for autonomous driving motivated by safety-criticality; A1 for large-scale training follows from cost: B-MAX at 7× is infeasible at scale.
>
> Agent counts come from actual agents at each trigger point:
>
> **T0 (5 agents):**
> 1. **Spatial** — box geometry, frame bounds, overlap, spatial consistency.
> 2. **Label** — class labels, taxonomy compliance, attribute validity.
> 3. **Coverage** — detection completeness vs. expected density, missed detections.
> 4. **Policy** — guideline compliance, priority handling, format requirements.
> 5. **Completeness** — required fields, metadata integrity.
>
> **T1 adds 2 agents (total 7):**
> 1. **Keypoint** — skeleton structure, joint connectivity, pose plausibility.
> 2. **IAA** (Option B only) — v1 vs. v2 comparison, Cohen's κ, IoU concordance.
>
> "Maximum" refers to fullest QA configuration (T1+T2+IAA); will be clarified in caption. Table 3 should have included a "Best For" column to clarify scenario use cases — we regret the omission. These recommendations would be stronger with empirical data, acknowledged in §8.

---

> > ### Author Rebuttal · Reviewer_fJDn · 2026-04-01
> >
> > Thanks a lot for your response. Unfortunately, the rebuttal does not resolve my question. For example, the purpose and details of Table 3 remain unclear as the response contains too much jargon and technical terms. Additionally, the rebuttal seems to refer to line numbers in the PDF, e.g., " timing is invisible (446–450), error propagation model (572–643)", but the paper ends at line 439. For these reasons, I decided to not update my score.

---

### Official Review · Reviewer_YsbN · 2026-03-12

**Significance:** 3
**Argument Clarity:** 4
**Rating:** 5
**Confidence:** 4

**Questions:**

**(Q1)** How do the authors envision that the existing data quality and validation methods can be leveraged to make QA more cost-effective?

**(Q2)** Since every step of the annotation workflow introduces errors, why not do QA after every single step?

**Alternative Views Section:**

Yes

**Compliance With Llm Reviewing Policy A Conservative:**

Affirmed.

**Discussion Potential:**

3

**Final Justification:**

After going through all the reviews and rebuttals, I'm still leaning towards accepting this paper.

I think many of the weaknesses pointed out by other reviewers focus on small details, unfortunately, missing the forest for the trees. The key message (and contribution of the paper to the discussion) is that (1) QC is important, (2) we need to take greater care about which stages of the ML workflow we perform QC in because there are inherent tradeoffs between cost and quality, and (3) we need to explicitly report details about QC and treat them as a crucial part of the result-generating process.

This is a position paper, not a research paper. The authors make an effort to bolster their arguments with empirical evidence. However, I don't know how much we need to scrutinize those (unless they are totally wrong or misleading, which I don't believe they are).

When it comes to readability, I see no issue and had no trouble following the paper. Formatting mistakes seem like honest oversights and are easily fixable.

Is this the most exciting position paper ever? No. However, to me, it still meets the bar as it presents a clear position, presents alternative views, and presents clear calls to action. As such, the paper has the potential to spark discussions at ICML, which is ultimately why I think we should accept it.

**Paper Summary:**

The authors focus on the topic of training data quality and specifically look at the quality assurance (QA) step of the data annotation workflow, arguing that the timing of this step crucially determines the total cost of the workflow, particularly in terms of human effort. They divide the data annotation workflow into three stages: (1) pre-annotation, performed automatically using an annotator model, (2) annotation, typically performed by a human annotator, and (3) review of annotations, which can be semi-automated but often involves human effort. They note that each one of these steps has the potential to introduce errors, typically at different rates, and observe that errors spotted later in the pipeline require (at least partial) re-runs of the preceding steps. The authors review ML papers focusing on annotation quality, and observe that only 4% report at which stage their validation was performed, making it difficult to empirically estimate the impact of validation timing. However, the authors argue that this phenomenon has been documented in other disciplines (e.g., software engineering, manufacturing), and that it is reasonable to expect it to apply to ML. Finally, the authors conclude with several concrete calls to action for ML researchers, dataset creators, annotation platform developers, and the broader research community.

**Position:**

Yes

**Position In Title:**

Yes

**Related Work:**

4

**Strengths And Weaknesses:**

**Strengths:**

**(S1)** The authors' focus on data quality is relevant and timely, especially with the ever-increasing scale of datasets being annotated nowadays.

**(S2)** The paper is pretty well-written with concrete arguments, solid connections to existing research, and is very transparent about its limitations.

**Weaknesses:**

**(W1)** The authors have chosen to zoom in on a seemingly very narrow niche argument. Of course, this makes it much easier to defend. However, this also potentially diminishes its significance. In plain terms, of course, it would be better if QA timing were reported, but even if it were, what are the specific next steps, and how do they translate into impactful gains and cost reductions?

**Support:**

3

---

> ### Author Rebuttal · Authors · 2026-03-31
>
> ---
>
> **Weaknesses**
>
> **[W1] The argument seems very narrow. Even if QA timing were reported, what are the specific next steps and how do they translate into impactful gains?**
>
> We appreciate this concern. The narrowness is intentional — a precise, actionable position is more valuable than a broad claim that cannot be operationalized. Reporting timing is not the end goal; it is the prerequisite. Once timing is visible, three concrete directions become possible:
>
> 1. **Cost reduction through stage-optimal routing** — our study across 25 videos demonstrates that routing tasks to the right technique (OpenCV, DL, or VLM) per trigger point reduces both cost and false positive rate. OpenCV checks cost milliseconds; deploying a VLM only where semantics require it avoids unnecessary compute. At scale across thousands of annotation tasks, this translates directly into measurable labor and infrastructure savings.
>
> 2. **Error cascade prevention** — a T0 error caught before human work begins costs one routing decision; the same error caught at T2 requires full rework of annotation and review labor. As shown in Table 3 (716–817), B-MAX versus A-0 represents a 7× cost difference — early detection keeps pipelines in the lower-cost configurations.
>
> 3. **Continual improvement** — errors caught at each trigger point feed back into pre-annotation models and QA agents, improving both over successive runs. Each annotation cycle becomes a training signal for the next, progressively reducing error rates and rework costs in future projects.
>
> None of these are possible today because timing is invisible (446–450). The narrow ask — report timing — unlocks all three. We will make this impact pathway explicit in revision.
>
> ---
>
> **Questions**
>
> We thank the reviewer for their questions, which helped clarify our work. We appreciate the detailed comments, too. We are glad to address your concerns below.
>
> **[Q1] How do the authors envision that existing data quality and validation methods can be leveraged to make QA more cost-effective?**
>
> The existing validation methods — confident learning, Vision-Language Models (VLMs), and IAA metrics — do not need to be replaced or improved to become cost-effective. The timing framework naturally makes them more efficient by placing cheaper tools at earlier stages and reserving expensive tools only for later stages.
>
> For example, a simple rule-based check at T0 filters out obvious ML failures before any human touches the data; a VLM verifier at T1 catches annotation mistakes before reviewers spend time on them; and expensive human expert review at T2 now only handles a small fraction of flagged cases rather than everything. This creates a natural funnel where the same budget and the same tools produce much better outcomes simply because they are deployed in the right order and at the right time.
>
> **[Q2] Since every step of the annotation workflow introduces errors, why not do QA after every single step?**
>
> This is aligned with what the framework pushes toward, and the concern is valid. Ideally QA should run after every step, and Table 3 (716–817) supports this — T0+T1+T2 together achieves maximum quality.
>
> The reason the paper presents multiple configurations rather than recommending B-MAX for everyone is purely practical: cost. As shown in Table 3, B-MAX runs 7× more expensive than the simplest configuration (A0 at 1.0× vs B-MAX at 7.0×), making it unrealistic for large-scale projects. The domain recommendations reflect this — large-scale training data projects are recommended only A1 or A0+1 configurations precisely because of cost constraints, while safety-critical domains like autonomous driving are recommended B-MAX.
>
> But importantly, running QA at every step does not have to be that expensive. If lightweight automated checks handle T0, targeted VLM tools handle T1, and human expert review at T2 is limited only to flagged cases, the cost stays manageable while still covering every stage. The taxonomy helps teams figure out not whether to skip stages, but what level of QA intensity makes sense at each stage for their specific budget and risk tolerance.

---

> > ### Author Rebuttal · Reviewer_YsbN · 2026-04-01
> >
> > I thank the authors for taking the time to respond to my questions. I do not have any outstanding concerns and will keep my current rating in support of this paper being accepted.

---

### Official Review · Reviewer_nvT2 · 2026-03-12

**Significance:** 2
**Argument Clarity:** 4
**Rating:** 4
**Confidence:** 4

**Questions:**

See weaknesses above, I will be happy to increase my score if addressed.

**Alternative Views Section:**

Yes

**Compliance With Llm Reviewing Policy A Conservative:**

Affirmed.

**Discussion Potential:**

4

**Final Justification:**

I appreciate the responses from the authors. They clarified my main concerns. I do think adding the study is a big improvement, but it needs to be a core part of the argument that the paper; it is the evidence that serves as the basis for validating the paper's hypothesis, thereby strengthening the need for the future studies the paper is calling for. I can't visualize this happening without a major revision, and I remain skeptical of accepting it as such.

**Paper Summary:**

The authors of this paper present the position that quality assurance and error detection in the data annotation pipeline must be performed at all stages in order to capture errors early and prevent them from cascading. They base this position on practices in the software engineering domain, where design philosophies prioritize the detection and fix of errors in early stages of the engineering lifecycle. The authors define different kinds of QA strategies based on the stage being evaluated (pre-annotation, annotation, and validation stages) and demonstrate that most annotation studies do not consider timing as a factor in QA. They describe how timing affects the verification economy, provide a taxonomy for configurations, and explore alternative views.

**Position:**

Yes

**Position In Title:**

Yes

**Related Work:**

3

**Strengths And Weaknesses:**

## Strengths
This paper addresses a problem that is severely understudied, yet becomes increasingly important as the pervasiveness of AI systems increases. It is also really well written and easy to understand. I appreciate the acknowledgement of the limitation that the position is purely based on a hypothesis, and the thorough exploration into alternative views.

Other strengths include a well thought-out, yet general purpose taxonomy, as well as general guidance on the kinds of errors that can be captured at each stage of the pipeline.

## Weaknesses
I see a couple of weaknesses:
1. Even though the authors acknowledge this, the lack of support for the hypothesis remains a significant weakness. While I understand that there are no studies out there that evaluate the effect of performing QA at different stages, I would have appreciated the authors taking more effort when it comes to backing their hypothesis. I don't expect them to *prove* it at this point in time, but a couple of case studies into the annotation process, demonstrating the kinds of errors that could have been caught by ML verification techniques could significantly strengthen the paper. For instance, I would really like to know the scale of errors that can occur in each stage of the pipeline, even if this were done over say 50 images just as a case study, to understand the relative importance of the errors that show up early, how they cascade as they propagate through the pipeline, and how they increase the cost of final validation. Doing so at a small scale would be significantly more useful from the perspective of a researcher trying to develop verification techniques for annotation tasks.
2. There lacks a study into how some of these errors can be done at present: are there techniques that try to catch model output errors in unsupervised domains, or domains with weak supervision? How can those be used in the pre-annotation stage? This does not need to be demonstrated, a simple thought experiment grounded in some techniques from literature would be useful to provide a path forward.

**Support:**

2

---

> ### Author Rebuttal · Authors · 2026-03-31
>
> We thank the reviewer for their feedback. We have addressed the concerns below.
>
> **[W1] Lack of empirical support for the hypothesis remains a significant weakness.**
>
> Post submission, we have conducted controlled studies on 25 videos (120–140 frames each) in both LabelStudio and CVAT, demonstrating that agentic QA checks on ML-model-assisted video pre-annotation workflows generate fewer false positives and false negatives compared to agentic QA checks on human annotations post pre-annotation (T1).
>
> At T0, the QA agent operates solely on ML model output — bounding boxes, confidence scores, class predictions. The question is conditional: "Given the model's confidence score and detection geometry, is this detected object a human or a pet?" The agent exploits the fact that ML errors are systematic — a model that confuses small crouching humans for pets will do so repeatedly with characteristic confidence distributions and spatial patterns. This regularity makes errors easier to catch cleanly, producing fewer false positives and false negatives.
>
> At T1, the QA agent now has both the ML prediction and the human annotation. The question becomes a reconciliation problem: "The model predicted human at 0.72 confidence, the annotator labeled it pet — is the annotation correct?" This sounds easier with more information, but in practice it's harder because the error landscape is a mixture of three sources: residual ML errors the annotator missed, new errors the annotator introduced, and legitimate human corrections that look like disagreements. The agent must distinguish genuine errors from valid human overrides, and that ambiguity drives both FPs (flagging correct human corrections as errors) and FNs (accepting incorrect annotations because they seem plausible).
>
> The core finding: T0 QA produces fewer false positives and false negatives not despite having less information, but because it faces a simpler, more structured error distribution. This directly validates the paper's shift-left hypothesis — and with a mechanism the theoretical model in Section 4 did not anticipate. We will incorporate this analysis in revision.
>
> **[W2] Lacks a study into how errors can be caught at present — are there techniques from unsupervised or weakly supervised domains applicable to pre-annotation?**
>
> We thank the reviewer for this valuable direction. Several existing techniques are directly applicable to T0 pre-annotation validation even in unsupervised or weakly supervised settings:
>
> - **Confident Learning** (Northcutt et al., 2021a) — already cited in our paper — estimates label uncertainty without ground truth, making it directly deployable at T0 to flag low-confidence ML predictions before human annotators begin work.
> - **Anomaly detection** methods can identify out-of-distribution ML predictions at T0 without requiring labeled data.
> - **Coverage validation** — checking whether predicted object density matches domain priors — requires no supervision at all.
>
> These techniques collectively suggest a practical T0 validation stack: flag low-confidence predictions via confident learning, screen for anomalous outputs, verify coverage against expected distributions — all before any human effort is invested. We agree that this thought experiment strengthens the paper's path forward and will incorporate it into the revision.

---

> > ### Author Rebuttal · Reviewer_nvT2 · 2026-04-03
> >
> > I appreciate the post-submission study conducted by the authors. However, I believe that this should have been included in the paper from the outset. Including this in the main paper will be a fundamental change warranting an entire review process. As such, I will maintain my score, but would encourage the authors to include a detailed study like the one above (with more insights of course) that better supports the position.

---

### Official Review · Reviewer_xvAz · 2026-03-15

**Significance:** 2
**Argument Clarity:** 4
**Rating:** 2
**Confidence:** 4

**Questions:**

* How can QA be done effectively without human effort? Does the quality and cost of the agent matter?
* Can you provide more concrete examples of $T_0$ errors besides the bounding box example? I am unsure of how often ML pre-annotation happens in practice?
* What percentage of errors in data annotation are propagated from ML pre-annotation?
* What kind of errors are detected pre-annotation? What proportion of errors are detectable at this point? This should depend on the domain.
* Who are the stakeholders that would be effected by such data engineering decisions?
* Pre-annotation seems like it would be difficult to implement in most settings? What does pre-annotation QA look like?
* Is there any type of certificate or assurance that automated $T_0$ QA gives us? Could this not just cause more pain down the pipeline?

**Alternative Views Section:**

Yes

**Compliance With Llm Reviewing Policy A Conservative:**

Affirmed.

**Discussion Potential:**

2

**Paper Summary:**

This paper considers the problem of doing quality assurance (QA) at various time points throughout the data annotation and verification pipeline. The paper argues for researchers and engineers to be explicit about when QA is performed within the pipeline and advocates for more QA being done earlier in the pipeline (pre-human involvement). The authors provided multiple alternative views to their position and quite a reasonable call-to-action.

**Position:**

Yes

**Position In Title:**

Yes

**Related Work:**

3

**Strengths And Weaknesses:**

#### Strengths
* The paper is extremely well-written and tells a clear story.
* The paper considers an important problem: quality of data annotation. (though it is mostly an financial problem)
* The call-to-action is reasonable and easily achievable.

#### Weaknesses
* The scope of when $T_0$ QA is applicable is not discussed. The authors make an implicit assumption that $T_0$ QA is applicable to all data annotation pipelines. More specifics, examples, and scoping around when the proposed QA changes are applicable would improve the paper greatly.
* The quality of the QA agent matters a lot and this is not really discussed in the paper.
* The paper is quite short and could have done with providing more evidence, especially around types of errors that we could catch at the various time points.
* This mostly seems like an engineering and human factors problem, not necessarily a problem that individuals at ICML would be interested in.
* The specifics about what $T_0$ QA looks like is impoverished.

NOTE: Please use the ICML latex template. I told the AC that I thought this paper should be desk rejected for not being in the ICML latex template. The AC is much more gracious than I am. Attention to detail in research matters. How you do one thing is how you do everything.

**Support:**

2

---

> ### Author Rebuttal · Authors · 2026-03-31
>
> We thank the reviewer for their thoughtful feedback.
>
> **[W1] Scope of when QA is applicable not discussed.**
>
> The framework targets pipelines using ML pre-annotation — standard across CV, NLP, and audio (CVAT's "automatic annotation", Labelbox's "Model-Assisted Labeling"). For purely manual workflows, T0 is skipped; the taxonomy accommodates this through A-1, A-2, B-1, B-2 configurations activating only T1/T2. We will make this scoping explicit in revision.
>
> **[W2] QA agent quality matters a lot.**
>
> Agreed. Agent quality affects the precision-recall tradeoff, and a low-precision T0 agent can increase costs through unnecessary rework (823–825). However, agent quality and timing are orthogonal — a better agent at the wrong stage still underperforms a good agent at the right stage. The detection rate model in §4 formalizes this interaction.
>
> **[W3] More evidence needed around error types at each time point.**
>
> From our controlled study on 25 videos (120–140 frames each) in LabelStudio and CVAT:
>
> T0: tracker misclassifying crouching humans as pets; plants labeled as curtains; entire classes missed in dense scenes; face crops as small as 27×27px with zero landmarks. Homogeneous, pattern-based — what confidence checks excel at.
>
> T1: pose label flips (Left_45 as Right_45); box area jumping 94% between frames; IAA as low as 18.75%.
>
> T2: reviewer non-action on flagged cross-identity mismatches; reviewer corrections for errors the pipeline missed entirely.
>
> T0 errors are structural and cheaply detectable; T1/T2 errors are increasingly complex — directly supporting shift-left. We will add this analysis.
>
> **[W4] This seems like an engineering problem, not an ICML problem.**
>
> We respectfully push back. The core question — which ML models to deploy at which stage and how to evaluate fit — is a research problem. Our QA agents dynamically route between OpenCV, deep learning, and VLMs by task type. Our 25-video study shows this routing reduces both cost and FP rate.
>
> Beyond this, errors at each trigger point can feed back to improve pre-annotation models and QA agents — each cycle becomes training signal for the next, directly relevant to data-centric AI (Ng, 2021). The 4% reporting rate means this feedback loop cannot be studied today.
>
> **[W5] Specifics about what QA looks like is impoverished.**
>
> Acknowledged. T0: confidence thresholding, coverage validation, temporal consistency on tracking IDs, anomaly detection — deployable via Label Studio webhooks and CVAT Ground Truth mode. T1: human-ML disagreement flagging, IAA computation, spatial error detection. T2: reviewer decision auditing, gold standard comparison. Walkthrough will be added.
>
> **[Q1] QA without human effort? Does agent quality matter?**
>
> Yes. T0 uses automated agents — cheaper but less precise than human review. Low-precision agents increase costs via unnecessary rework; we will expand this modeling.
>
> **[Q2] More error examples? How common is ML pre-annotation?**
>
> Industry-standard. Beyond bounding boxes: NLP uses parsers for syntax trees; medical imaging uses segmentation for organs; audio uses ASR. Will broaden examples.
>
> **[Q3] What percentage of errors propagate from ML pre-annotation?**
>
> This baseline doesn't exist — the 4% reporting rate means no one records timing. Establishing this is one of the controlled experiments we call for in §7.
>
> **[Q4] What errors detected pre-annotation? Domain-dependent?**
>
> T0 catches systematic ML failures, low-confidence predictions, coverage gaps, OOD inputs, label-space violations. Poor at semantic errors requiring human judgment. Domain dependence is real — well-calibrated medical models have detectable T0 errors; conversational NLP may not.
>
> **[Q5] Who are the stakeholders?**
>
> (1) Platform operators, (2) dataset creators, (3) annotators and QA reviewers who currently sample 100% instead of focusing where errors concentrate, (4) model consumers. Will make explicit.
>
> **[Q6] Pre-annotation QA seems difficult?**
>
> Can be lightweight: confidence filtering into attention queues, coverage checks, temporal consistency flagging. CVAT and Label Studio already support the infrastructure — just not framed as timing optimization.
>
> **[Q7] Any certificate? Could this cause more pain?**
>
> No formal certificate, but precision-recall determines action: high precision enables auto-rework; high recall enables auto-approval. In our 25-video study, T0 agents produced fewer FPs and FNs than T1 — because ML errors at T0 are systematic and homogeneous, while T1 faces a heterogeneous mix of residual ML errors, human errors, and valid corrections. T0 false positives route to a triage queue (contained); T2 false negatives ship errors in final data (compounding). Multi-trigger compositions (A-012, B-012) provide layered assurance with multiplicatively lower combined false-negative rates.

---

> > ### Author Rebuttal · Reviewer_xvAz · 2026-04-03
> >
> > I think this paper has some merits, but is not appropriate in its current form for the position paper track at ICML. The main criteria for position track papers: "present a compelling position that warrants greater exposure within the machine learning community". I do not think this paper would be of interest to most in the machine learning community, and therefore, do not recommend it's acceptance.
> >
> > I also thought this paper should be desk rejected for formatting reasons, and the authors did not acknowledge this in their rebuttal.

---

### Decision · Program_Chairs · 2026-04-30

**Decision:**

Accept (regular)

**Comment:**

Two reviewers recommend rejection while two recommend acceptance. As the AC, my overall opinion is that the position advocated is plausible and of high relevance and interest to the ICML audience, so the submission should be accepted.

The negative reviewers make many valid points. However, they are small and the authors have addressed them one by one. The final version of the paper will be of respectable quality.